# Theoretical and Experimental Models to Evaluate the Possibility of Corrosion Resistant Concrete for Coastal Offshore Structures

**DOI:** 10.3390/ma15134697

**Published:** 2022-07-04

**Authors:** Sergey Fedosov, Boris Bulgakov, Hung Xuan Ngo, Olga Aleksandrova, Vadim Solovev

**Affiliations:** 1Department of Building Materials Science, Moscow State University of Civil Engineering, Yaroslavskoe Shosse, 26, Moscow 129337, Russia; fedosovsv@mgsu.ru (S.F.); xuanhung1610@gmail.com (H.X.N.); aleksandrovaov@mgsu.ru (O.A.); solovevvg@mgsu.ru (V.S.); 2Department Building Design, Institute of Construction and Architecture, Volga State University of Technology, Lenin Square, 3, Yoshkar-Ola 424000, Russia; 3Faculty of Civil Engineering, Hanoi University of Mining and Geology, No.18 Vien Street, Duc Thang Ward, Bac Tu Liem District, Hanoi 11910, Vietnam

**Keywords:** reinforced concrete structure, concrete corrosion, non-stationary mass transfer, Fourier numbers, free calcium hydroxide, mathematical model

## Abstract

This study built theoretical and practical models to evaluate the corrosion resistance of concrete for coastal offshore structures in Vietnam. A mathematical model was developed in the form of a system of nonlinear partial differential equations characterizing the diffusion “free calcium hydroxide” in a solid of a concrete structure. The model describes the process of non-stationary mass conductivity observed in the “concrete structure—marine environment” system under non-uniform arbitrary initial conditions, as well as combined boundary conditions of the second and third kind, taking into account the nonlinear nature of the coefficients of mass conductivity *k* and mass transfer *β*. It was shown that the solution of the boundary value problem of non-stationary mass conductivity allows us to conclude about the duration of the service life of a concrete structure, which will be determined by the processes occurring at the interface: in concrete—mass conductivity, depending on the structural and mechanical characteristics of hydraulic structures, and in the liquid phase—mass transfer, determined by the conditions of interaction at the interface of the indicated phases.

## 1. Introduction

The South China Sea plays an important role in Vietnam’s history, since the coastline from north to south is about 3260 km. Many important economic centers and defense facilities of the country are located on the coast. The accumulated experience shows that many hydrotechnical reinforced concrete structures, after 5 on 10 years of operation, develop damage caused by corrosion processes occurring in an aggressive marine environment. The rate of corrosion damage is quite high, especially in tidal waters. Thus, the issues of increasing the reliability and durability of the operation of hydrotechnical facilities in the coastal zone of Vietnam are very relevant and are of great economic and social importance for the country [1].

The destruction of reinforced concrete structures occurs due to corrosion processes caused by diffusion (mass transfer) between the concrete components and the ions of aggressive components of the liquid phase [2,3,4]. Recent studies only stop at building a model to evaluate the process of chloride ion penetration from the aggressive environment into reinforced concrete structures [5,6,7,8]. There have been no specific studies on the corrosion process caused by diffusion between the concrete components.

This paper considers the basic methods of physical-mathematical modeling that are used to describe the processes of non-stationary mass transfer of “free calcium hydroxide” in concrete structures placed in a liquid environment with a defined flow rate. The boundary value problem of “free calcium hydroxide” mass conductivity in dimensionless variables is obtained. To demonstrate the possibilities of the obtained solution, we will carry out a numerical experiment: in which the fluctuation in the field of dimensionless concentrations $C(\bar{x},Fo_{m})$ by the different values of the Fourier number, in accordance with the theory of similarity, will be considered as an indicator of the processing time. The study indicated the results of calculating the concentration distributions “free calcium hydroxide” over the dimensionless thickness of the concrete structure at Fourier numbers 0.01; 0.1; 0.2; 0.5 and 1. The study also provides an example of determining the time of reaching the critical concentration “of free calcium hydroxide” on the coastal structure surface.

Theoretical model is applied to corrosion resistant concrete (CRC) with a modified structure based on sulfate resistant Portland cement using mainly local materials, suitable for the construction of offshore structures in coastal areas, which can be carried out by compaction and strengthening of the structure of the cement stone due to the combined effect of modifying admixtures (MA), introduced into the concrete mixture, in the form of a water-reducing polycarboxylate superplasticizer (*SP*), as well as silica fume (*SF*) and mechanically activated low-calcium fuel fly ash (*FA*) and rice husk ash (*RHA*)—finely dispersed mineral admixtures the composition of a multicomponent adhesives and having a high pozzolanic activity due to a significant content of amorphous silica [9,10].

## 2. Materials and Methods

Sulphate resisting portland cement type CEM I 42.5N CC (*SC*) produced by the Tam Diep plant, which is the leading cement manufacturer in Vietnam using the most modern world technologies. The main characteristics of clinker and Portland cement based on it met the requirements of ASTM C150-07 [11], GOST 22266-2013 [12] (State standard of Russia) and TCVN 6067:2018 [13] (State standard of Vietnam).

The physical and mechanical properties, as well as the chemical and mineral compositions of the used cement are shown in Table 1, Table 2 and Table 3.

Active mineral admixtures allow to reduce the consumption of cement, as well as to compact the structure of concrete by reducing the porosity of the cement stone and thereby improving its operational properties, and in addition, to avoid stratification of the concrete mixture when using water-reducing superplasticizers [14,15,16]. Local active mineral ingredients used in the work included fly ash class F from Vung Ang (*FA*) conforms to the standard TCVN 10302:2014 [17] and GOST 25818-2017 [18], Vina Pacific SF-90 Silica fume (*SF*) and rice husk ash (*RHA*) conforms to the standard TCVN 8827:2011 [19]. Their composition and properties are shown in Table 4 and Table 5.

Granulometric composition of *FA*, *SF* and *RHA*, shown in Figure 1, was determined using the method of laser granulometry.

Silica sand (*SS*) of the Lo River (Vietnam) was used as a fine aggregate. It is a popular construction sand in Vietnam with good quality and low price. The grain size composition of sand is important for the preparation of concrete mixtures of the required consistency, since it has a significant effect on their workability and the amount of mixing water required for this. The regulatory requirements for the physical and mechanical properties of sand are set out in the Russia and Vietnam standards GOST 8736-2014 [20] and TCVN 7570:2006 [21]. The results of their determination are presented in Table 6.

As a coarse aggregate, we used crushed stone (*CS*) with D_max_ = 10 mm, which is mined in open pits in Ninh Binh (Vietnam) and whose properties corresponded to the requirements of the standards GOST 8267-93 [22] and TCVN 7570:2006 [21]. The physical and mechanical properties of the used crushed stone are shown in Table 7.

A special requirement is imposed on the cleanliness of the aggregate, since dusty, silty and clay particles envelop the surface of the grains and impair their adhesion to the cement stone. Therefore, the content of such particles in a coarse aggregate should not exceed 3%.

The superplasticizer SR 5000P (*SP*) from SilkRoad (Vietnam) with a density of 1.1 g/m^3^ at a temperature of 20 ± 5 °C was used as a plasticizing additive in concrete mixtures, which reduces the water demand of equally mobile concrete mixtures by 30–40% that meets the requirements of GOST 24211-2008 [23] and ASTM C494/C494M-19 [24]. The main characteristics are shown in Table 8.

According to the passport data provided by the manufacturer, the optimal dosage of the superplasticizer SR5000P for obtaining a concrete mixture with the highest mobility is in the range of 0.9 ÷ 1.2% of the mass of the adhesives. If the SP consumption exceeds this amount, then this can lead to water separation and stratification of the concrete mixture. Therefore, the work used the average value of the recommended dosage of the superplasticizer in the amount of 1% by weight of the adhesives.

Water (*W*) used for the preparation of concrete mixtures complied with the requirements of GOST 23732-2011 [25] and TCVN 4506:2012 [26]. Such water should not contain impurities that affect the setting of concrete, as well as reduce the durability of structures, above the permissible limit, have a pH value of at least 4 and contain no more than 5.6 g/L of mineral salts, including no more than 2.7 g/L sulfates. In addition, the water should be free of sludge and oil flakes, as well as organic matter of more than 15 mg/L.


**Building theoretical models**


Sea water is a highly corrosive environment containing a large amount of dissolved salts and causing chemical corrosion of both concrete itself and steel reinforcement in reinforced concrete structures. The aggressive marine environment has a significant impact on the durability of concrete and reinforced concrete structures of hydraulic structures of the coastal zone. At the same time, in reinforced concrete, the penetration of liquid aggressive media through capillary pores causes cracking and peeling of the protective concrete layer above the surface of the reinforcing bars, which leads to corrosion of the reinforcement [27,28,29,30].

To experimentally determine the chemical composition of seawater at different depths in the coastal zone, in the area of Halong port in the north of Vietnam, samples were taken (Figure 2), the results of chemical analysis of which are presented in Table 9.

Table 9 shows that the content of solutes in seawater tends to increase in the bottom layer, especially the content of *Ca*^2+^ ions. This is due to the fact that Halong Bay rests on a limestone base, as a result of which the seawater of the bottom layer, dissolving calcium-containing rocks, has a higher concentration of *Ca*^2+^ ions.

For the most part, all offshore hydraulic structures are made of concrete or reinforced concrete, complex composite materials, the viability, performance, and durability of which to a decisive extent depend on the structure of structures, their physicochemical, structural, mechanical, and operational properties. An important influence is exerted by the salinity of sea water, the presence of salts of inorganic substances in it and the presence of biological microorganisms in different climatic seasons. From the point of view of the theories of physicochemical hydrodynamics and heat and mass transfer, the nature of the interaction of the composite of a hydraulic structure with the components of seawater is determined by the laws of chemical kinetics and diffusion in the bulk of concrete and at the solid-liquid interface, as well as by the laws of mass transfer (in this case, the transfer substances from the interface into the volume of the sea water basin).

To develop effective methods for protecting concrete from leaching by a marine environment containing a range of different ingredients that have a significant effect on the rate of decomposition of highly basic compounds and the removal of decomposition products into the marine environment, it is necessary to develop mathematical models of unsteady mass conductivity (diffusion in a solid) under non-uniform arbitrary initial conditions and combined boundary conditions of the 2nd and 3rd kind. Particular attention should be paid to taking into account the nonlinearity of the coefficients of mass conductivity and mass transfer.

In accordance with the classification of Professor V.M. Moskvin [31], the simplest form of development of corrosion processes in concrete is leaching. In this case, the aggressive component does not penetrate deep into the material of the concrete (reinforced concrete) structure. The rate of the process is determined by the diffusion of calcium hydroxide from the pores of the inner layers of the structure to the external solid-liquid interface, and then by mass transfer from the interface to the liquid mass.

In this case, it is assumed that the target component, which is free calcium hydroxide in the processes of corrosion of cement concrete, is removed from the surface of a concrete or reinforced concrete structure by a liquid medium as a result of convective mass transfer. If the medium is stationary, then the mass transfer will be characterized by natural convection, and if the surface of the structure is washed with a liquid at a certain speed of its movement, then there is a forced flow of the liquid. In both cases, the mass transfer of the target component will be determined by two processes: mass conductivity from the inner layers to the interface and mass transfer from the interface to the liquid phase [32,33,34,35,36,37,38,39].

The model of the problem of mass transfer with initial and boundary conditions for an unbounded plate concrete (reinforced concrete) can be schematically illustrated in Figure 3.

The problem of mass transfer of calcium hydroxide from a concrete structure into an aqueous substance can be formulated by the following system of Equations (1)–(4):(1)$$\frac{\partial C(x,\tau )}{\partial \tau }=k\frac{\partial^{2}c(x,\tau )}{\partial x^{2}},\tau >0,0\leq x\leq \delta ,$$
(2)$$C(x,0)=C_{0},$$
(3)$$\frac{\partial C(0,\tau )}{\partial x}=0,$$
(4)$$\beta \left[C\left(\delta ,\tau \right)-C_{p}\right]=-k\frac{\partial C(\delta ,\tau )}{\partial x},$$
where: *C*_0_ is the initial concentration of free calcium hydroxide in concrete, in terms of calcium oxide, kg CaO/kg concrete; *C(x,τ)* is the concentration of free calcium hydroxide in concrete at the moment *τ* at any point with the coordinate *x*, in terms of calcium oxide, kg CaO/kg concrete; *k* is coefficient of mass conductivity in the solid phase (diffusion), m^2^/s; *β* is mass transfer coefficient in a liquid medium, m/s; *Cp* is the equilibrium concentration of the transferred component on the surface of a solid; kg CaO/kg concrete; *δ* is wall thickness of the structure, m.

The Equation (1) is the differential equation of non-stationary mass transfer in the body of a reinforced concrete structure. The Equation (2) defines the initial condition of the process: the distribution of calcium hydroxide concentrations at the time instant taken as the initial one. The Equations (3) and (4) expressions define the conditions at the interface. The Equation (3), called the condition of the 2nd kind, also called the “non-penetration condition”, determines the fact that calcium hydroxide does not diffuse into the internal premises of the hydraulic structure located on the left from enclosing concrete (reinforced concrete) construction. The Equation (4) characterizes the interaction of the surface layer of the structure with a liquid medium. This is a condition of the 3rd kind, also called “Newton’s condition”.

The use of dimensionless variables allows you to go to the following Equation (5):(5)$$C\left(\bar{x},Fo_{m}\right)=\frac{C\left(x,\tau \right)-C_{p}}{C_{0}-C_{p}},\;\bar{x}=\frac{x}{\delta },\;Fo_{m}=\frac{k\tau }{\delta^{2}},\;Bi_{m}=\frac{\beta \delta }{k},$$
where: $C\left(\bar{x},Fo_{m}\right)$ is the dimensionless concentration of the transferred component across the concrete thickness; $\bar{x}$ is dimensionless coordinate; $Fo_{m}$ is Fourier mass transfer criterion; $Bi_{m}$ is Bio mass transfer criterion.

In this case, the system of Equations (1)–(4), also called the “boundary value problem of non-stationary mass transfer”, is transformed to the from:(6)$$\frac{\partial C\left(\bar{x},Fo_{m}\right)}{\partial Fo_{m}}=\frac{\partial^{2}C\left(\bar{x},Fo_{m}\right)}{\partial {\bar{x}}^{2}},\;Fo_{m}>0,\;0\leq \bar{x}\leq 1,$$
(7)$$C\left(\bar{x},0\right)=1,$$
(8)$$\frac{\partial C\left(0,Fo_{m}\right)}{\partial \bar{x}}=0,$$
(9)$$\frac{\partial C(1,Fo_{m})}{\partial \bar{x}}=-Bi_{m}C(1,Fo_{m}),$$

The purpose of solving this boundary value problem is to find a function $C\left(\bar{x},Fo_{m}\right)$ that allows one to calculate the concentration profiles of the transferred component over the thickness of the structure, which also change over time. This is the so-called “direct problem of the dynamics of the mass transfer process” [40]. The solution to the abovementioned problems is indicated in the [32,41].
(10)$$C\left(\bar{x},Fo_{m}\right)={2Bi}_{m}\sum_{m=1}^{\infty }\frac{\cos\left(\mu_{m}\bar{x}\right).\exp(-\mu_{m}^{2}{\mathrm{Fo}}_{m})}{\mu_{m}\left[\sin\left(\mu_{m}\right)\left(1+Bi_{m}\right)+\mu_{m}.\cos\left(\mu_{m}\right)\right]},$$
where $\mu_{m}$ is the roots of the characteristic Equation (11):(11)$$tg(\mu_{m})=\frac{Bi_{m}}{\mu_{m}},\;\mathrm{or}\;ctg(\mu_{m})=\frac{\mu_{m}}{Bi_{m}},$$

Some results of calculations by Equation (10) are shown in Figure 4.

## 3. Results

Based on the studied causes and nature of corrosion of offshore hydraulic structures, in order to increase the durability of cement stone of concrete and reinforced concrete structures, experimental studies were carried out in the laboratories of the Civil Engineering Faculty of the Hanoi Mining and Geological University and the Institute of Construction Sciences and Technologies of the Ministry of Construction of Vietnam in accordance with the requirements of the standards ACI 211.4R-08 [42], developed corrosion resistant concrete, the compositions of which are shown in Table 10.

### 3.1. Investigation of the Physical and Mechanical Properties and Performance Indicators of the Developed Corrosion Resistant Concrete

The experimental results of determining the physical, mechanical, and performance indicators of the developed concretes of the above compositions and, for comparison, the requirements for concrete in accordance with SP 41.13330.2012 [43] are presented in Table 11.

From the test results given in Table 11, it can be seen that an increase in the density of concrete of Mix 3 and 4 due to the use of fine mineral fillers in the form of silica fume and mechanically activated ash of rice husks containing amorphous silica capable of binding free calcium hydroxide (CH) into less soluble low-basic calcium hydrosilicate (CSH), contributes not only to an increase in strength, but also to an increase in the water-resistance of concrete, as well as a decrease in water absorption. At the same time, the concrete of the Mix 3 has the highest compressive strength in comparison with the rest of the developed concretes. This is due to the increased content of CSH formed as a result of the pozzolanic reaction due to the high content of *SiO*_2_ (89.9%) in the silica fume of the mineral sealing additive, which is confirmed by the results of X-ray phase analysis.

These results are fully consistent with the results of the analysis of the microstructure of concrete of the developed compositions, obtained using the method of electron microscopy and showing a denser structure in CRC of Mix 3 and 4 compared to Mix 1 and 2 (Figure 5).

### 3.2. Study of the Effect of Finely Dispersed Active Mineral Admixtures on the Composition of Hydration Products by X-ray Phase Analysis

In order to assess the pozzolanic properties of the used mineral admixtures (*FA*, *SF,* and *RHA*), the method of X-ray phase analysis was used to study their influence on the phase composition of the adhesive’s hydration products during the hardening of concrete of Mix 1, 2, 3 and 4.

The obtained results of studying the influence of these active mineral admixtures on the change in the phase composition of hydrated neoplasms in the cement stone of concrete of the developed compositions at the age of 28 days of hardening are presented in Figure 6.

The presented figures show that, in contrast to Mix 1 and 2, the intensity of the peaks of free *Ca*(*OH*)_2_—portlandite decreases in concretes of Mix 3 and 4, and at the same time, the intensity of the peaks of calcium hydrosilicate increases. This can be explained by the occurrence of the pozzolanic reaction of *SF*, *RHA,* and *FA* with portlandite, the rate of which increases with the hardening of these concretes, as a result of which the absorption of calcium hydroxide and its transformation into hydrosilicate occurs more intensively than the formation of *Ca*(*OH*)_2_ as a result of hydration of Portland cement clinker minerals. It was found that at the age of 28 days of normal hardening, the highest intensity of CSH peaks and the smallest CH peaks are observed in the concrete of Mix 3 containing silica fume, which can be explained by its high pozzolanic activity.

The results obtained allow us to conclude that mechanically activated ash and silica fume has a positive effect on the formation of low-basic calcium hydrosilicate in the structure of hardening concretes, which will increase their density, strength, and corrosion resistance under operating conditions.

### 3.3. Determination of the Coefficient of Mass Conductivity of Calcium Hydroxide by the Thickness of the Concrete Structure and the Forecast of the Duration of Operation of Concrete and Reinforced Concrete Structures

Determination of the coefficient of mass conductivity is not limited to purely technological problems associated with the fact that the coefficient of mass conductivity is included in the calculated equations of the ongoing processes, but is of great scientific importance, since allows you to study the mechanism of the process and the influence of various factors on the rate of transfer of matter [44].

To determine the content of calcium hydroxide over the thickness of the concrete structure, an experimental scheme was developed, as shown in Figure 7. For the tests, concrete samples with dimensions of 100 × 100 × 100 mm were used. The surfaces of the samples, except for one, were protected with a waterproof paint coating, leaving one face unprotected to allow the diffusion of calcium hydroxide into a bath with an aqueous solution simulating the composition of seawater in the bottom layer of the South China Sea in the area of Halong port. The specified solution contained chlorides of calcium, magnesium, and potassium, as well as sodium sulfate with a total concentration of calcium ions equal to 0.7 g/L, chlorine ions 18.2 g/L, sodium ions 11.3 g/L, potassium ions 0.4 g/L, magnesium ions 1.4 g/L and sulfate anions 2.7 g/L.

The content of calcium hydroxide was determined by thermogravimetric analysis in the central zone of the samples every 25 mm of thickness with a 14-day interval during 70 days of testing. As a result, profiles of calcium hydroxide concentrations were obtained over the thickness of the sample in an aqueous medium (Figure 8).

Analyzing the concentration profiles of calcium hydroxide over the thickness of concrete samples of different composition, we determined the concentration gradients of calcium hydroxide at the interface and, using Equation (1) and the Matlab and Origin 2018 software, calculated the value of the mass conductivity coefficient *k* of calcium hydroxide. The calculation results are shown in Table 12.

The change of mass conductivity coefficient with time is shown in Figure 9. From the obtained results, it can be seen that the mass conductivity coefficient decreases sharply in the period from 14 to 42 days. From 42 days to 70 days, the mass conductivity coefficient continued to decrease but not significantly.

As an illustration, the following is a specific example of calculating the corrosion time of a concrete structure:Thickness of the concrete structure of the hydraulic structure: *δ* = 0.3 m;Coefficient of mass conductivity of calcium hydroxide in concrete k according to Table 12 at the moment of time *τ* = 56 days.

Calculations using the proposed method in the Equation (1) show that the critical value of the dimensionless concentration of the transferred component, free calcium hydroxide, over the thickness of concrete $C\left(\bar{x},Fo_{m}\right)$ is achieved at the value of the mass transfer Fourier number *Fo_m,crit_*, equal to one (curve 6 in Figure 4). In accordance with the accepted designations, the calculation of the corrosion time will be carried out according to the Equation (12):(12)$$\tau_{crit}=\frac{\delta^{2}Fo_{m,crit}}{k},$$

The results of calculating the corrosion time of concrete structures from the four investigated concrete compositions are shown in Table 13.

## 4. Discussion

To solve the problem of protecting a reinforced concrete structure from the aggressive effects of the marine environment, it is necessary to use the obtained expression to solve the “inverse problem of non-stationary mass transfer” in order to find conditions under which the processes of mass transfer would be carried out with a minimum leaching rate. It is possible to control this process by influencing the structure of the concrete in the structure. Obviously, the time parameter is the Fourier mass transfer criterion. It is also obvious that the time *τ* is included in the exponential function, which is a factor in each term of the Fourier series. Therefore, the solution to the “inverse mass transfer problem” is possible only with the use of the iteration method [41].

From the results of calculations given in Table 12 and Table 13, it can be seen that the mass conductivity coefficient of calcium hydroxide in the concrete of Mix 3 is less than that of the control concrete of Mix 1. Consequently, the concrete structure from the Mix 3 will have a longer service life. This result is explained by the fact that concrete of composition No. 3 has a dense structure due to the simultaneous use of mechanically activated *FA* in the composition of multicomponent adhesives and *SF* as an active mineral finely dispersed sealing additive. The research results show that the developed experimental model can be used to solve the inverse problem of unsteady mass conductivity in order to determine the mass conductivity coefficient of calcium hydroxide in a concrete structure. This model can serve as a basis for predicting the service life of the concrete and reinforced concrete structures of hydraulic structures in the marine aquatic environment.

## 5. Conclusions

(1) Compositions of corrosion resistant concretes have been developed for the construction of underwater parts of offshore hydraulic structures of the coastal zone, the structure of which is modified with active mineral admixtures (including silica fume, mechanically activated rice husk ash, and fly ash) and polycarboxylate superplasticizer in the form of water.

(2) A mathematical model has been developed in the form of nonlinear partial differential equations characterizing unsteady mass conductivity—diffusion in a solid of a concrete (reinforced concrete) structure of free calcium hydroxide, observed in the system “concrete (reinforced concrete) structure—marine environment” under uneven arbitrary initial conditions, as well as combined boundary conditions of the 2nd and 3rd kind, taking into account the nonlinear nature of the coefficients of mass conductivity *k* and mass transfer *β*.

(3) It is shown that the solution of the boundary value problem of unsteady mass conductivity allows us to conclude about the durability of the concrete (reinforced concrete) structure, which will be determined by the processes occurring at the interface: in concrete—mass conductivity, depending on the structural and mechanical characteristics of hydraulic structures, and in liquid phase—mass transfer, determined by the conditions of interaction at the interface between these phases. At the same time, the analysis of the liquid phase will make it possible to assess the duration of the serviceability of these structures and, as a result, it becomes possible to design the optimal compositions of corrosion resistant concretes intended for the construction of durable offshore structures, due to their high resistance to corrosion in seawater.

## Figures and Tables

**Figure 1 materials-15-04697-f001:**
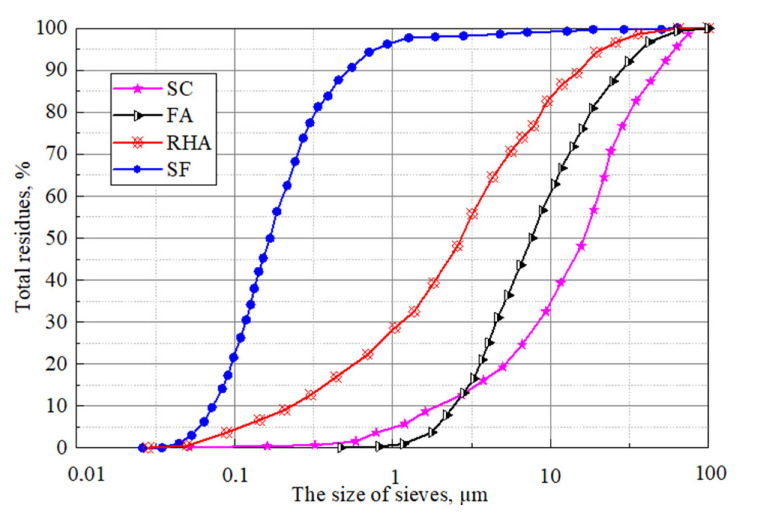
Granulometric composition of *SC*, *FA*, *SF* and *RHA*.

**Figure 2 materials-15-04697-f002:**
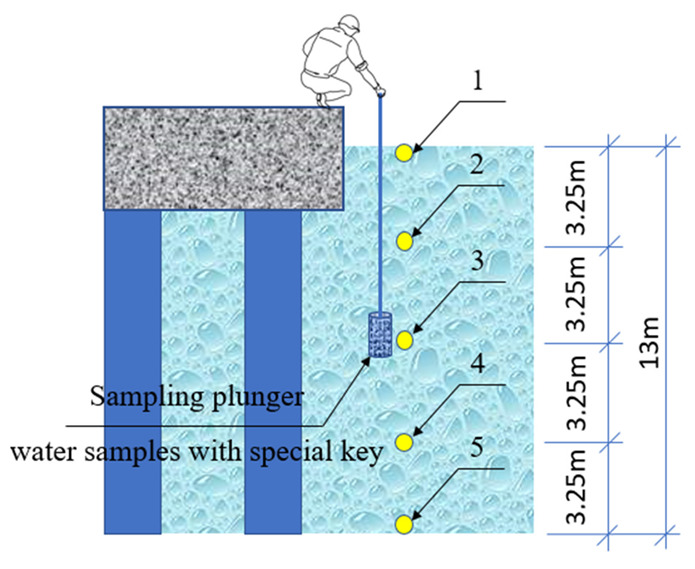
Scheme of seawater sampling to determine the chemical composition in the Halong port area (North Vietnam): 1, 2, 3, 4, 5—water sampling points.

**Figure 3 materials-15-04697-f003:**
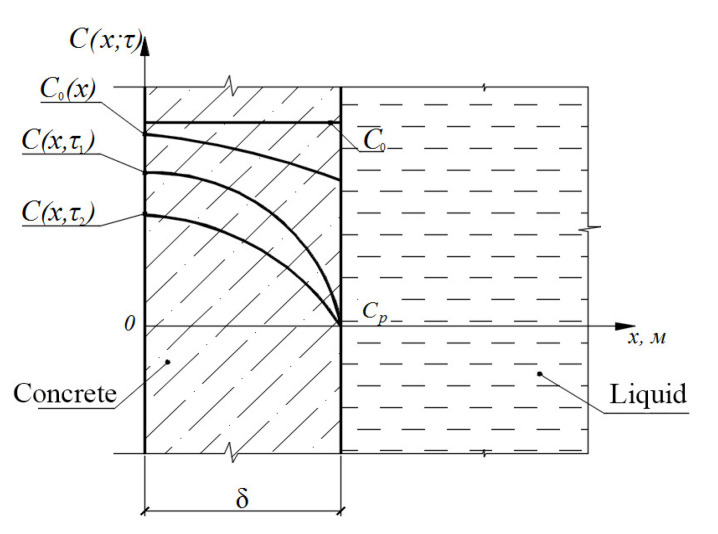
On the physical and mathematical formulation of the problem.

**Figure 4 materials-15-04697-f004:**
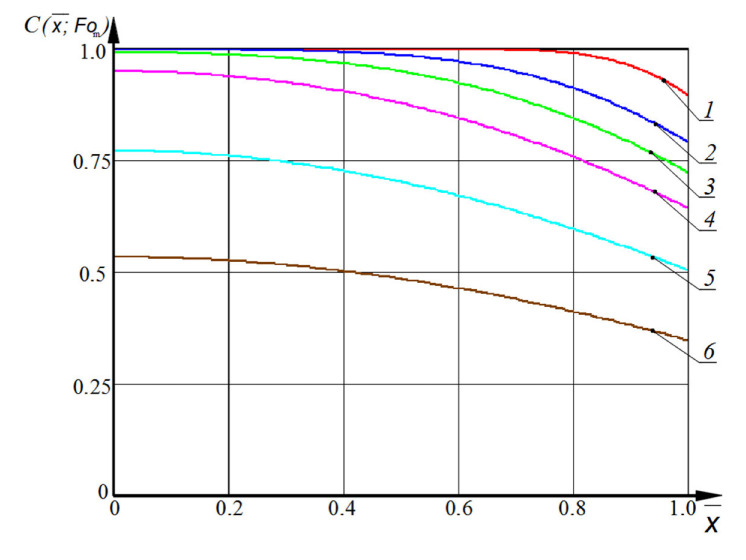
The profiles of dimensionless concentrations $C(\bar{x},Fo_{m})$ over the thickness of the concrete structure at Fourier numbers: *Bi_m_* = 1, *Fo_m_* = 1—0.01; 2—0.05; 3—0.1; 4—0.2; 5—0.5; 6—1.

**Figure 5 materials-15-04697-f005:**
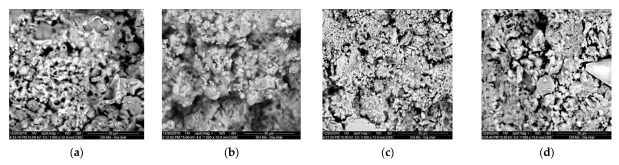
The microstructure of the developed concrete at the age of 28 days of normal hardening (increase ×6000): (**a**) Mix 1; (**b**) Mix 2; (**c**) Mix 3; (**d**) Mix 4.

**Figure 6 materials-15-04697-f006:**
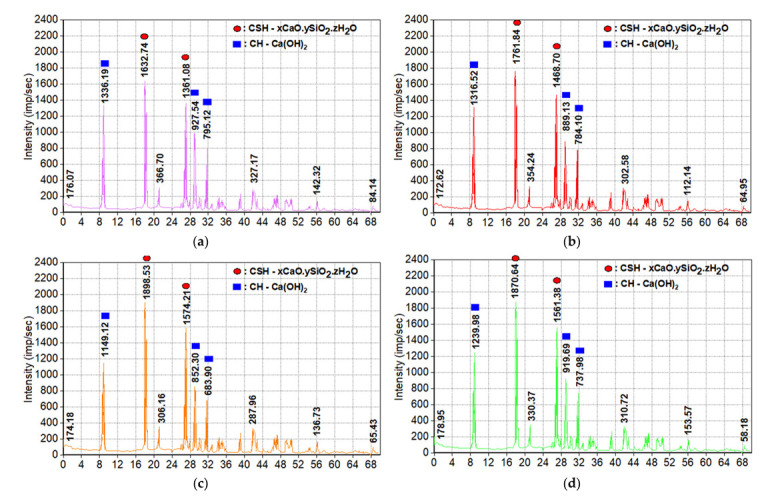
X-ray diffraction pattern of samples of compositions: (**a**) Mix 1; (**b**) Mix 2; (**c**) Mix 3; (**d**) Mix 4.

**Figure 7 materials-15-04697-f007:**
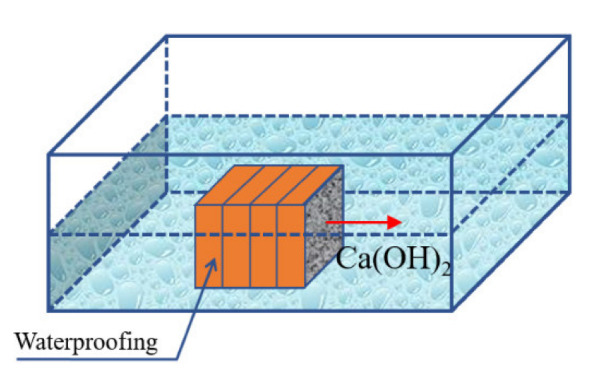
Scheme for determining the mass conductivity coefficient of calcium hydroxide over the thickness of a concrete structure.

**Figure 8 materials-15-04697-f008:**
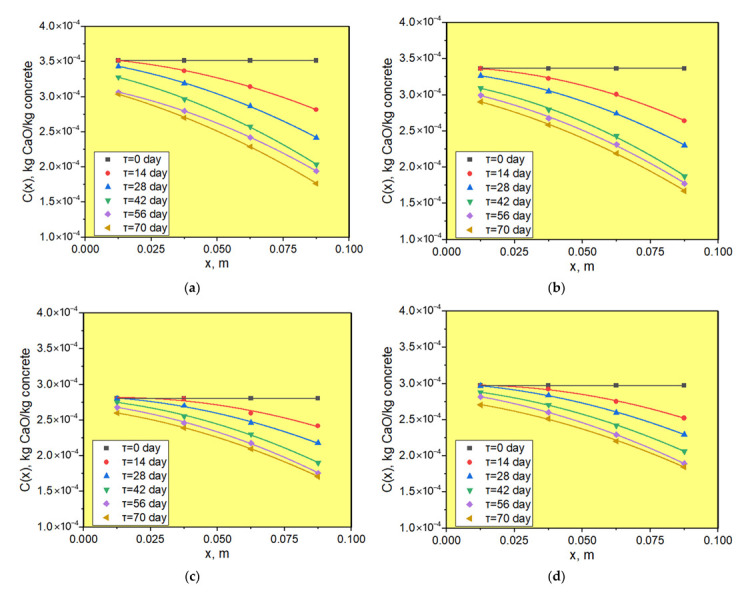
Profiles of *Ca*(*OH*)_2_ concentrations over the thickness of samples: (**a**) Mix 1; (**b**) Mix 2; (**c**) Mix 3; (**d**) Mix 4.

**Figure 9 materials-15-04697-f009:**
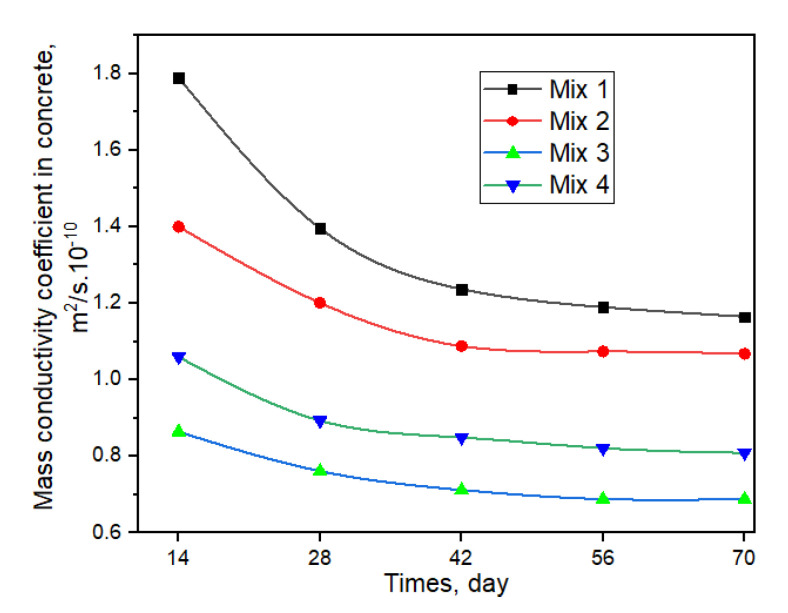
Effect of mass conductivity coefficient on time.

**Table 1 materials-15-04697-t001:** Physical and mechanical properties of sulphate resisting portland cement.

Properties	Result	Properties	Result
Water consumption (%)	28.1	Residue on the 90 µm sieve (%)	3.7
2-day compressive strength (MPa)	25.6	Blaine specific surface (cm^2^/g)	3631
28-day compressive strength (MPa)	51.3	Volume stability (mm)	1.2
Initial setting time (min)	162	Specific density (g/cm^3^)	3.12
Final setting time (min)	285	Average diameter of cement grains (µm)	28.92

**Table 2 materials-15-04697-t002:** Chemical composition of sulphate resisting portland cement.

Chemical Composition, %								
*SiO* * _2_ *	*A1* *_2_O* * _3_ *	*CaO*	*Fe* *_2_O* * _3_ *	*MgO*	*SO* * _3_ *	*Na* *_2_O*	*K* *_2_O*	L.O.I *
21.97	3.68	64.65	4.55	1.95	1.76	0.16	0.3	0.97

* Loss on ignition.

**Table 3 materials-15-04697-t003:** Mineral composition of sulphate resisting portland cement.

Mineral Content, %				
*C* *_3_S*	*C* *_2_S*	*C* *_3_A*	*C* *_4_FA*	Other
62.8	21.3	2.1	11.5	2.3

**Table 4 materials-15-04697-t004:** Chemical composition of active mineral admixtures.

Name of Additive	Chemical Composition, %										
	*SiO_2_*	*Al_2_O_3_*	*CaO*	*Fe_2_O_3_*	*MgO*	*SO_3_*	*Na_2_O*	*K_2_O*	*TiO_2_*	*P_2_O_5_*	L.O.I *
*FA*	54.62	25.17	2.65	7.11	1.57	0.25	0.95	2.18	1.83	0.63	3.04
*SF*	89.9	1.0	0.48	1.0	0.85	0.95	1.55	2.12	-	-	2.15
*RHA*	82.2	5.25	2.52	1.75	0.8	0.5	2.67	1.14	0.15	0.25	2.77

* Loss on ignition.

**Table 5 materials-15-04697-t005:** Physical properties of the used active mineral admixtures.

Active Mineral Admixtures	Density, (g/cm^3^)	Bulk Density, (g/cm^3^)	Average Particle Size, (µm)	Specific Surface, (m^2^/g)
*FA* *	2.22	0.51	2.746	16.725
*SF*	2.25	0.45	0.282	25.218
*RHA* *	2.31	0.57	2.854	14.480

* The characteristics of *FA* and *RHA* mechanically activated by grinding in a vibrating ball mill for 20 min are given.

**Table 6 materials-15-04697-t006:** Basic physical and mechanical properties of silica sand.

Size Module, M_K_	Density, (g/cm^3^)	Bulk Density, (g/cm^3^)	Humidity, (%)
2.95	2.65	1.24	0.8

**Table 7 materials-15-04697-t007:** Physical and mechanical properties of crushed stone.

Density, (g/cm^3^)	Bulk Density in Compacted State, (kg/dm^3^)	Humidity, (%)
2.75	1.6	0.4

**Table 8 materials-15-04697-t008:** Main characteristics of the superplasticizer.

Density at 25 °C, (g/cm^3^)	Hydrogen Exponent, (pH)	Alkalinity, (%)	Dosage, (% wt. of Adhesives)
1.1	6.0 ÷ 7.5	20 ÷ 26.5	0.9 ÷ 1.2

**Table 9 materials-15-04697-t009:** Composition of seawater in the South China Sea at different depths off the coast of North Vietnam in the Halong port area.

Water Sampling Locations	Indicators of Aggressiveness						
	pH	*Na* ^+^	*K* ^+^	*Ca* ^2+^	$Cl^{-}$	$SO_{4}^{2-}$	*Mg* ^2+^
		(g/L)	(g/L)	(g/L)	(g/L)	(g/L)	(g/L)
1	8.4	10.8	0.3	0.4	18	2.5	1.2
2	8.4	10.8	0.3	0.3	18.1	2.4	1.2
3	8.5	10.9	0.4	0.4	18.1	2.4	1.3
4	8.5	11.2	0.4	0.5	18.2	2.6	1.3
5	8.7	11.3	0.4	0.7	18.2	2.7	1.4

**Table 10 materials-15-04697-t010:** Developed compositions of corrosion resistant concrete.

Compositions	*W/AD* ^1^	Consumption of Raw Materials per 1 m^3^ (kg)							
		*SC*	*FA*	*SF*	*RHA*	*SA*	*CS*	*SP*	*W*
Mix 1	0.32	545	-	-	-	536	1040	5.5	174
Mix 2	0.32	474	71	-	-	536	1040	5.5	174
Mix 3	0.32	474	71	47	-	536	1040	5.9	189
Mix 4	0.32	474	71	-	47	536	1040	5.9	189

^1^ Loss on ignition *AD* = *SC* + *FA* + *SF*; *W/AD* = 0.32 was determined as a result of optimization of the compositions of concrete mixtures by the method of mathematical planning of the experiment using computer programs Matlab and Maple-2019.

**Table 11 materials-15-04697-t011:** Technological properties of concrete mixtures and physical and mechanical properties and performance indicators of the developed concretes.

Indicators		SP 41.13330.2012	Concrete Mixes and Concretes			
			Mix 1	Mix 2	Mix 3	Mix 4
Concrete mix mobility, cm		-	18	15	16	17
Average density of concrete mix, kg/m^3^		-	2282	2278	2347	2345
Concrete density, g/cm^3^		-	2441	2425	2492	2487
Average density of concrete, kg/m^3^		2330	2268	2252	2329	2316
Compressive strength, MPa	28 days	B5-B40 (less than 6.5–51.4 MPa)	62.9	60.6	78.5	76.8
	180 days	-	63.3	61.1	79.3	77.6
Axial tensile strength, MPa	28 days	B_t_0.8–B_t_3.2 (less than 1.03–4.11 MPa)	3.5	3.2	4.3	4.1
	180 days	-	3.6	3.3	4.5	4.2
Water absorption,% wt.		-	3.4	3.6	2.3	2.7
Total pore volume,%		-	7.09	7.13	6.54	6.88
Filtration coefficient, cm/s		6.6 × 10^−13^	2.1 × 10^−12^	2.1 × 10^−12^	4.2 × 10^−13^	4.8 × 10^−13^
Waterproof grade		W2–W20	W10	W8	W16	W16

**Table 12 materials-15-04697-t012:** Experimentally calculated characteristics of mass transfer of calcium hydroxide in concrete samples.

Concrete	Mass Conductivity Coefficient in Concrete (m^2^/s∙10^−10^)				
	*τ*, Day.				
	14	28	42	56	70
Mix 1	1.790	1.396	1.237	1.190	1.165
Mix 2	1.400	1.201	1.088	1.074	1.068
Mix 3	0.864	0.761	0.712	0.688	0.688
Mix 4	1.060	0.893	0.849	0.821	0.809

**Table 13 materials-15-04697-t013:** Results of calculating the corrosion time of concrete structures.

Indicators	Mix 1	Mix 2	Mix 3	Mix 4
Corrosion time (s.)	7562 × 10^8^	8381 × 10^8^	1308 × 10^9^	1096 × 10^9^
Corrosion time (y.)	24	27	41	35

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
