# Peer review of "Theoretical and Experimental Models to Evaluate the Possibility of Corrosion Resistant Concrete for Coastal Offshore Structures"

_materials, 2022, doi:10.3390/ma15134697_

Round 1

Reviewer 1 Report

The authors adopt the theoretical model of calcium ion precipitation for the research and development of anti-corrosion concrete mix ratio. This work is interesting. The article generally meets the requirements of Materials, but there are some issues that must be revised before it can be considered for acceptance.
1. Line 30-32, "The South China Sea plays an important role in Vietnam's history, since the coastline from north to south is about 3260 km and includes more than 3000 islands and archipelagos." The author should be responsible for the above text. As far as the reviewers know, Vietnam has very few sovereign islands in the South China Sea, and most of the islands belong to China.
2. Line 34, "after 5 10 years" symbol should be changed to to
3. Although the authors improved the corrosion resistance of concrete by means of modification, their chloride ion attack ability was not considered in this study. Do the concrete mix ratios provided by the author help to improve the impermeability of concrete?
4. Figure 12 and Table, please give the detailed reasons for the change of mass transfer coefficient

Author Response

  1. Information about islands in the South China Sea belonging to Vietnam has been deleted.
  2. The symbol in the phrase "after 5-10 years" has been replaced with the preposition "on".
  3. In Table 11, information about the waterproofness of the developed concretes has been added in the form of an established brand for waterproofness and values of the filtration coefficient.
  4. The reasons for changing the values of the mass transfer coefficient given in Table 12 are indicated in the second paragraph "Discussion".

Reviewer 2 Report

Dear Authors,

The corrosion mechanism is not explained and incorporated with a suitable SEM image and, along with the schematic diagram, explains all the possibilities with a clear idea of the corrosion science behind your research work.
Many characterizations are missing and identifying the mechanical and physical properties is required for your research work. It should be implemented with the suggested property evaluation.
Please clearly include the error bar analysis in all charts.

(1) Significance: 
Corrosion mechanism is not explained with the addition of an SEM image.

(2) Quality of Presentation: 
Many characterization and experimental validation are required further to improve the quality of the manuscript.
the data and analyses are not presented appropriately.
Significantly more improvements are required with the characterization of property study

(3) Scientific Soundness: 
It is highly modified after incorporating mechanism and property improvement in their studies.

Please address the comments carefully in the revised manuscript for further consideration of their work and do the needful in this regard.

Author Response

  1. The physical and mechanical properties of the developed concretes are presented in Table 11.
  2. We believe that the mechanism of corrosion of concrete cement stone due to leaching of free calcium hydroxide is widely known and for this reason does not require a separate description. The images of the microstructure of the developed concretes obtained by electron microscopy and presented in Figure 5 are given to confirm the established fact that the inclusion of a multicomponent binder in the composition of sulfate-resistant micro-silica Portland cement or mechanically activated rice husk ash, which have high pozzolanic activity due to the content of a large amount of amorphous silica in them, leads to compaction of the concrete structure due to the binding of free calcium hydroxide into low-base hydrosilicates.

Reviewer 3 Report

This manuscript described the theoretical and practical models to evaluate the corrosion resistance of concrete for coastal offshore structures. It shows the strength of performing together a theoretical and experimental approach to solve a problem. This approach can make it possible to design the optimal compositions of corrosion-resistant concretes intended for the construction of durable offshore structures. This manuscript will be certainly suitable for publication after some minor revisions.

  1. Table 1: the standard deviation should be added to the results.
  2. Please add the name of the chemicals from Table 2-5 somewhere. 

Author Response

  1. Table 1 presents the results of an experimental determination of the physico-mechanical properties of the used sulfate-resistant Portland cement, carried out according to the current standard methods.
  2. The transcription of the abbreviations FA, SF and RHA given in Table 4 and 5 is given in the text in the paragraph before Table 4. We consider it inappropriate to give the names of oxides whose chemical formulas are given in Table 2 and 4, as well as the well-known basic minerals of Portland cement clinker (alite C3S, belite C2S, tricalcium aluminate C3A and four-calcium aluminoferrite C4AF).

Round 2

Reviewer 1 Report

The author has revised the comments and this article is acceptable. But some spelling mistakes need to be checked carefully before publication

Reviewer 2 Report

Dear Authors,

I suggest to the authors to incorporate more characterization of the mechanical properties in the future work.

Reviewer 3 Report

The authors did significant improvements to this manuscript. This manuscript can be accepted in its present form.